**Data Availability Statement:** All relevant data are within the paper and its Supporting Information files.

# Acute effects of the FIFA11+ and Football+ warm-ups on motor performance. A crossover randomized controlled trial

**Mojtaba Asgari**[ORCID]*, **Marcus Schmidt**[ORCID]℗, **Benedikt Terschluse**℗, **Maximilian Sueck**℗, **Thomas Jaitner**[ORCID]℗

Institute for Sport and Sport Science, TU Dortmund University, Dortmund, Germany

℗ These authors contributed equally to this work.
* mojtaba.asgari@tu-dortmund.de

## Abstract

### Introduction

Few studies including contradictory results have addressed the acute effects of the 11+ on motor performance, indicating a potentially reduced applicability of the program for warming up before competitions. This study aims to compare the acute effects of a soccer-specific warm-up (Football+) and the 11+ on motor performance.

### Materials and methods

Thirty-eight volunteer collegiate players (22 males; age = 21.1±1.9 years, height = 1.81± 0.06 m, weight = 73.4± 9.5 kg; 16 females; age = 21.3±1.5 years; height = 1.71± 0.07 m, weight = 67.8± 8.5 kg) underwent the 11+ and the Football+ in a randomized crossover design with a one-week washout. The Football+ starts with a self-estimated 40–50 percent running, followed by dynamic stretching of the hip muscles, shoulder contact, controlled lunge, Copenhagen exercise, and modified Nordic hamstring exercise. The second part involves roughly intensive small-sided games, followed by plyometric and anaerobic exercises in the third part. The warm ups' effects on performance were determined by a linear sprinting test (20 m), countermovement jump performance (CMJ), Illinois agility (IA), and dribbling speed (DS) tests. Within-subject differences were reported as the means and SD. Pairwise t tests at the significance level of $p<0.05$ were used to calculate the significant differences.

### Results

Overall, except for the CMJ (mean = -0.43±3.20 cm, p = 0.21, d = -0.13), significant differences for the 20 m sprint (mean = 0.04±0.10 s, p = 0.005, d = 0.42), IA (mean = 0.65±0.45 s, p = 0.01, d = 1.43), and DS (mean = 0.60±1.58 s, p = 0.012, d = 0.38) were observed. In females, significant differences observed only for IA (mean difference = 0.52±0.42 s, p<0.001, d = 1.24) and DS (mean difference = 1.29±1,77 s, p = 0.005, d = 0.73), with the Football+ showing superiority. In males, significant differences were found only for 20 m

**Funding:** The authors received no specific funding for this work.

**Competing interests:** The authors have declared that no competing interests exist.

sprinting (mean difference = 0.06±0.09, p = 0.005, d = 0.60) and IA (mean difference = 0.74 ±0.46, p<0.001, d = 1.62), with the Football+ having superiority.

## Discussion

Although practicable for injury prevention, the 11+ may not optimize acute performance and prepare players for high-intensity physical tasks as well as a well-structured, roughly intensive warm-up. Further gender-specific studies should evaluate the long-term effects of the Football+ on performance and injury prevention.

## Introduction

Primarily designed as an injury prevention-oriented warm-up, the 11+ has been proven to reduce overall, hamstring, groin, and knee injury incidences by 40%, 66%, 48%, 46%, respectively [1–6]. Meanwhile, one recent systematic review on the 11+ has demonstrated that an abundant number of studies have also addressed the side effects of the 11+ on biomechanical measures and performance parameters [7]. The outcomes affirm that mid- to long-term application of the 11+ reveals positive effects on biomechanical measures such as concentric and eccentric strength of the hamstrings, concentric strength of the quadriceps, core strength and stability, and balance [7]. Nevertheless, contradictory results have been reported for performance parameters such as agility, sprinting, vertical jump and change of direction [8–14].

Regarding the acute effects of the 11+ on performance, however, the literature is tightly limited, sparse, and inconsistent. To date, only four studies including small samples (N<20) have been published [15–18], three of which have investigated common maneuvers in soccer, such as agility, sprinting, and vertical jumping, among amateur male players. Cloak et al. (2014) stated that the 11+ has no impact on agility and vertical jump [18], whereas Bizzini et al. (2013) showed that the 11+ might optimize agility and sprinting [16]. More recently, Ayala et al. (2017) reported that, compared to a dynamic warm-up, the 11+ may even reduce acute sprinting in adult male and female amateur players and highlighted that soccer-specific dynamic warm-ups are more beneficial for player preparation for high-intensity performance [15]. This is the only study that has investigated acute effects of the 11+ on female players. However, given the limited sample and that the results were not presented for females separately, the outcomes might not be generalized. As a result, a serious lack of gender-specific knowledge exists regarding the level of technical ability and physical performance in general and immediately after the application of 11+ [19], though research has shown that the physiological, metabolic and anthropometric requirements of male and female soccer players are too similar [20].

It is widely accepted that a practical warm-up should optimize motor performance. Several studies have addressed the advantages of dynamic warm-ups on acute performance and concluded that high-intensity warm-ups involving small-sided games and anaerobic exercises result in superior performance in intermittent-sprint running, reactive agility, countermovement jump, and 20-m sprinting [21–23]. In this concept, the 11+ must demonstrate its ability to optimize the common performance measures in soccer, such as sprinting, jumping, dribbling, and agility, so that it could be promoted at the level of routine warm-ups currently used in amateur football.

The weak effects of the 11+ on acute performance might result from a low intensity and suboptimal sequencing of the exercises embedded in this program. The 11+ starts with jogging

exercises, dynamic stretches and controlled contacts followed by 10-minute strengthening exercises and ends with 2-min anaerobic exercises. Such a sequence is highly acknowledged for injury prevention [14] but may not properly prepare players for further intensive loads. Furthermore, known as a purely dynamic sport, soccer is characterized by a variety of anaerobic tasks, and warm-up modalities, as the last phase of athletes' preparation before high-intensity performance, are required to not only prevent injuries but also optimize players' performance acutely [24].

Taken together, the acute effects of the 11+ on performance and consequently its applicability for warming up before competitions and matches remain questionable. This is a prominent challenge considering that the acute effects of the 11+ on performance highly interact with the compliance and implementation of the program, which have shown a strong correlation to the success of each injury prevention protocol and to team success [25]. Additionally, only a pilot study including a small sample and potentially underpowered results has investigated the acute effects of the 11+ on female players [15]. However, that study did not compare male and female players, and whether they react differently to the 11+ remains questionable. Therefore, the primary aim of this study is to compare the acute effects of the 11+ and a newly developed football-specific dynamic warm-up (Football+) on sprinting, agility, vertical jump, and dribbling speed among collegiate players. The secondary aim is to determine any possible differences in males and females following application of the warm-up modalities. To develop the Football+, we extended the 11+ with more dynamic, soccer-oriented, and anaerobic exercises with the intention of preserving the positive effects on injury prevention and ensuring better preparation for subsequent intensive movement tasks using the available literature [22, 26–28]. Given the soccer-specific and dynamic nature of the Football+ program, we assume that it results in superior acute performance compared to the 11+.

## Methods

### Study design

The current study was conducted in accordance to the Helsinki declaration guidelines. The ethics committee of TU Dortmund University read and approved the study protocol. Participation was voluntary, and before study commencement, all participants signed a written informed consent letter. In a randomized crossover design, participants conducted two warm-up modalities with a 1-week washout phase in between. Following the application of each warm-up, participants' performance was tested using a standardized test battery for performance assessment. To reach the best allocation concealment and avoid the learning effect from occasion to occasion, an ABBA approach was applied [29], implying that half of the subjects began with the 11+, while the other half started with a dynamic warm-up. To eliminate the time slot between the warm ups and the tests, we randomly divided the participants into four subgroups in accordance with the ABBA approach guidelines [29] and performed the interventions and measurements in two consecutive sessions (9–10 and 10–11 AM) in a day. With that setup, the players underwent the test battery immediately after completion of the warm-ups. In the following week, the interventions and the test order were swapped. Two certified research assistants with two years of the 11+ experience were responsible for conducting the warm-ups and test battery. The test battery comprised counter movement jump (CMJ), linear sprinting, Illinois agility (IA), and dribbling speed (DS) tests and was run immediately after the warm-ups. Prior to the measurements, all players participated in two familiarization sessions. The measurements were conducted in the mornings under comparable conditions (sunny weather, 21–22˚Celsius, 55–60% humidity) on the artificial turf where the players performed their routine training sessions. The players were asked to not perform heavy physical

activities 24 h before measurements, sleep no later than 12 pm the night before measurements and drink only caffeine-free liquids 4 h before measurements. They were also asked to put on the normal soccer shoes that are regularly used in artificial turf.

## Inclusion/exclusion criteria

This study included collegiate students who were actively playing football. Participants had to be at least 18 years old and participate in at least two training sessions per week. Those with a history of injury within the last four weeks were excluded. Participants were also excluded if they missed a measurement session, were engaged in intensive physical activities 24 hours before the test session, fell ill or contracted an infection during the study period.

## Participants

G*Power software [30] calculated a sample of 34 for an expected effect size of 0.5, α error probability of 0.05 and power of 0.80 [31]. Indeed, forty-two collegiate players voluntarily agreed to attend the study and signed a written informed consent form. Four participants dropped out during the study period due to coronavirus infection. Thus, twenty-two male (age = 21.1±1.9 years; height = 1.81± 0.06 m, weight = 73.4± 9.5 kg) and 16 female (age = 21.3±1.5 years; height = 1.71± 0.07 m, weight = 67.8± 8.5 kg) players completed both warm-ups and the test battery and were included for statistical analyses.

## Interventions

Both interventions consist of three parts lasting 25 minutes but differ in content. The 11+ aims to tackle modifiable injury risk factors such as neuromuscular control, static and dynamic balance, and the hamstring/quadriceps strength ratio [32]. The first part includes 8 minutes of running exercises at low speed combined with active stretching and controlled contact with a partner. The running course consists of six to ten pairs of cones depending on the number of players, approximately five to six meters apart (length and width). The second part involves six different sets of exercises, including strength, balance, and jumping, lasting 10 minutes, followed by 2 minutes of speed running combined with football-specific movements and sudden changes in direction in the last part.

The 11+ exercises were described in detail by Soligard et al. [32]. The Football+ begins with a 2-min self-estimated running across the pitch at approximately 40–50% of maximal pace, followed by dynamic stretch of the hip (hip in/out, hip abduction/adduction, hip flexion/extension, backward and side running, shoulder contact and landing, controlled lunges, two dynamic core stability exercises, Copenhagen and a modified Nordic Hamstring Exercises. The second part includes 5-min roughly intensive small-sided games, namely, active passing, unanticipated dribbling tasks, and one vs one, followed by plyometric and anaerobic exercises in the third part lasting 5 min (see appendix one).

## Performance tests

The warm ups' effects on performance were measured through a linear 5-, 10-, and 20-m sprint (Fig 1) [8, 10], CMJ (Fig 2) [8, 9], IA (Fig 3) [8, 33], and DS (Fig 4) [34]. Each subject performed each test twice, and the best results were considered for data analysis.

For linear sprinting, the players stood a meter behind the baseline and tried to run through the path with a maximal pace. The IA is performed in a rectangle of 10×5 m. The players start in a prone position, run toward the barrier at 10 m at maximum speed, return, and perform a zigzag run around four barriers, each 3.3 m apart. The test ends with another straight run to

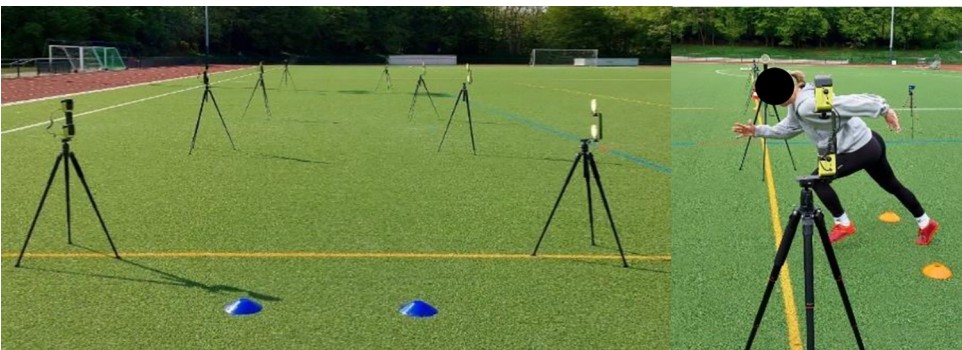

**Fig 1. Sprinting test.**

the end of the rectangle [33]. For the DS, the players start running with ball 3 m, then dribble the ball in a zigzag shape around three poles with a distance of half a meter each, then run with ball another 3 m and dribble the ball in a zigzag shape around last three poles with a distance of half a meter each. The test ends when the players pass the finish line. The CMJ is performed in a standing position on the mat and attempts to jump as high as possible. The flight time measure (Optojump Next System Microgate, Bolzano, Italy) measured the jump height. All tests were conducted in self-command start mode so that each player freely started the tests without receiving any external signal/command. Double-light time gates (90–110 cm height) were used to accurately measure the linear sprinting times at 5, 10 and 20 m distances as well as the time for completing the IA and DS [35].

## Statistical analyses

Descriptive measures, such as the means, SD and mean standard error (SE), were calculated for each outcome measure. Normal distribution of the data was proven for all variables using the Shapiro–Wilks test. Subsequently, the within-subject effects of the warm ups were analyzed by paired sample t tests. The level of significance was set at $\alpha < .05$. Magnitudes of differences were assessed using Cohen´s d effect sizes and interpreted as small (.25), medium (.5), and large (1.0) [36]. All statistical analyses were performed using SPSS version 28.

## Results

Table 1 presents the mean values, standard deviation (SD) and SE for the 11+ and the Football + (n = 38). As we expected that the results of the 5-, 10- and 20-m sprints were highly correlated, Pearson correlation coefficients were calculated for those variables. Due to the high values of r = 0.90–0.99 (p = 0.001), only the outcomes of the 20-m sprinting are presented in the results section.

Except for the CMJ (mean difference = -0.43±3.20 cm, p = 0.21, d = -0.13), the pairwise t test analyses revealed significant differences for the 20-m sprint (mean difference = 0.09±0.10 s, p = 0.005, d = 0.42), IA (mean difference = 0.65±0.45 s, p = 0.01, d = 1.43), and DS (mean difference = 0.60±1.58 s, p = 0.012, d = 0.38).

According to gender, males outperformed females across all parameters regardless of the warm ups (p≤0.005). In females, although differences were observed in 20 m sprinting and CMJ, the pairwise t test revealed significant differences only for IA (mean difference = 0.52 ±0.42 s, p<0.001, d = 1.24) and DS (mean difference = 1.29±1,77 s, p = 0.005, d = 0.73), with the Football+ showing superiority. In males, although differences were observed in DS, the statistical analysis revealed significant differences only for 20 m sprinting (mean difference = 0.06

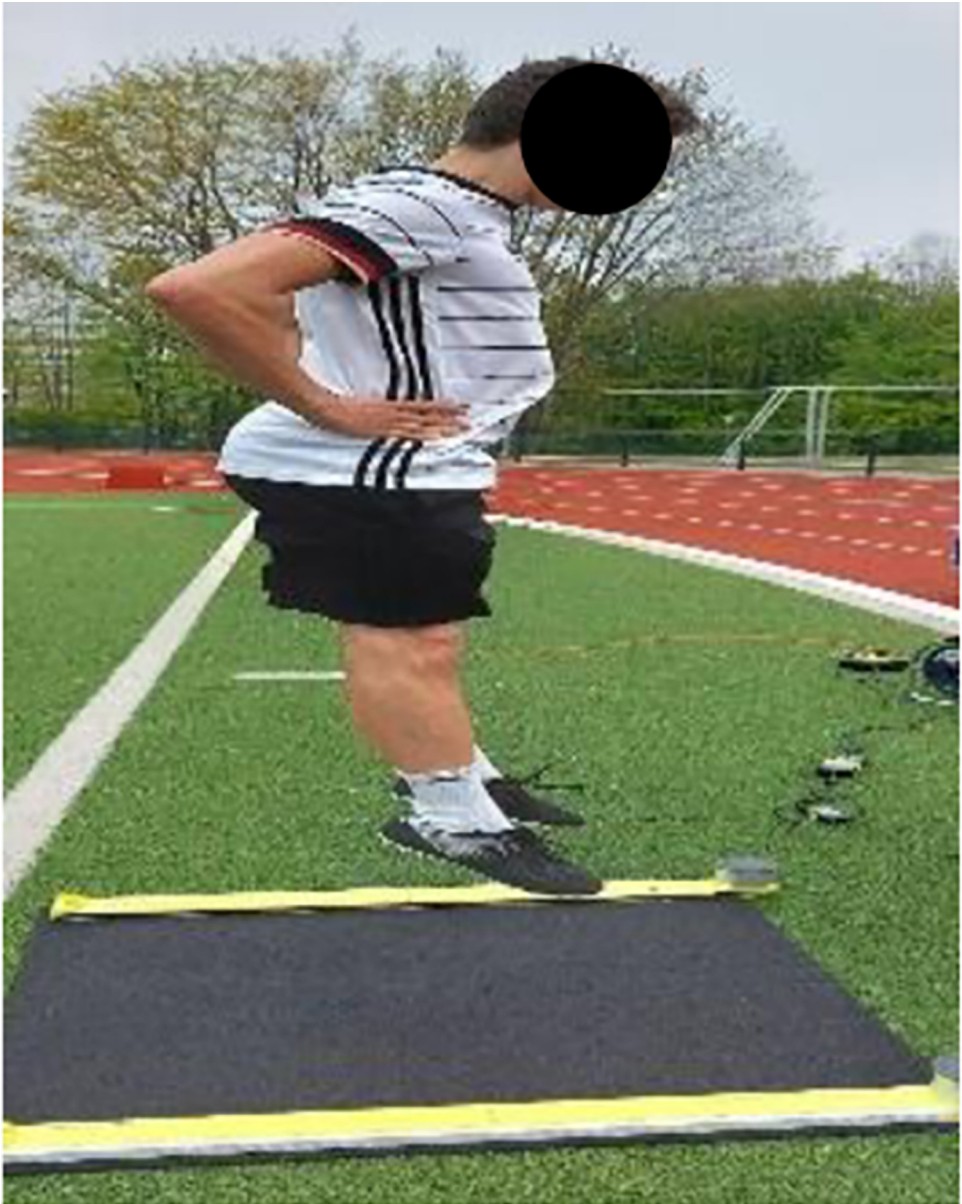

**Fig 2. Countermovement jump test.**

±0.09, p = 0.005, d = 0.60) and IA (mean difference = 0.74±0.46, p<0.001, d = 1.62), with the Football+ having superiority.

## Discussion

The primary aim of this study was to compare the acute effects of the 11+ and Football+ on motor performance and address concerns regarding the use of the 11+ as an appropriate warm-up modality before high-intensity training and competitions. In general, the findings demonstrate that the Football+ program appropriately optimizes acute performance, leading to superior operation in sprinting, agility, and dribbling, but not in the vertical jump, compared to the 11+ program in collegiate players. Our outcomes strongly support Ayala et al.

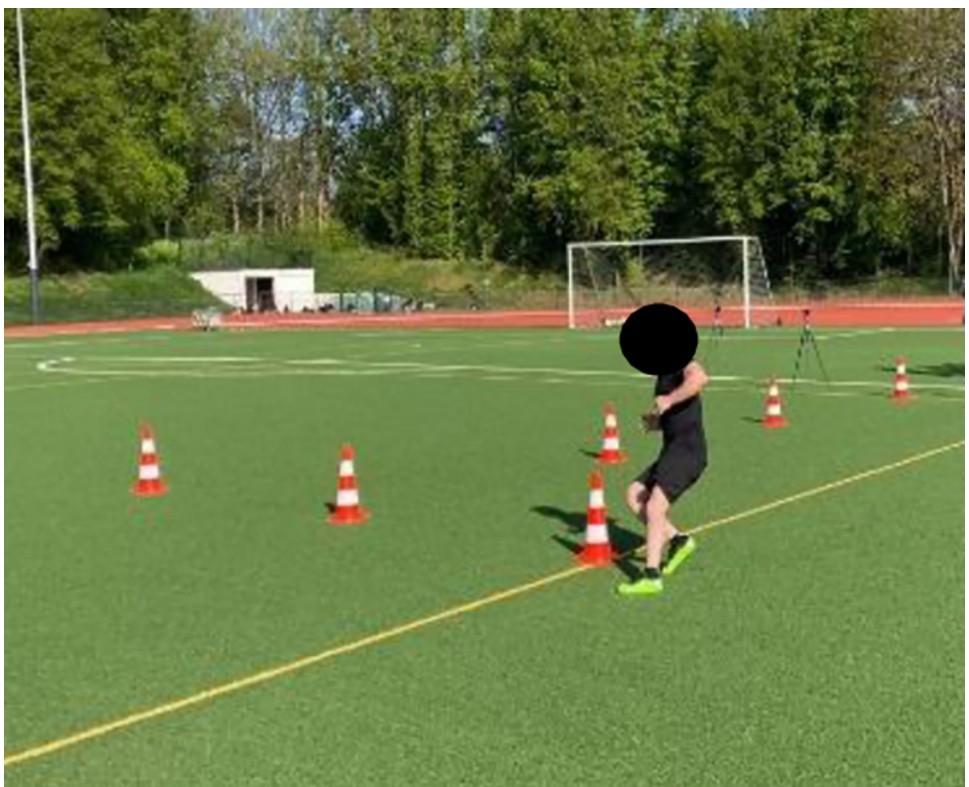

**Fig 3. Illinois agility test.**

(2017), who found no acute impact on sprinting or vertical jump following the application of the 11+ compared to a dynamic warm-up among amateur male and female soccer players [15]. Interestingly, given the identical study design and the mixed sample being undertaken,

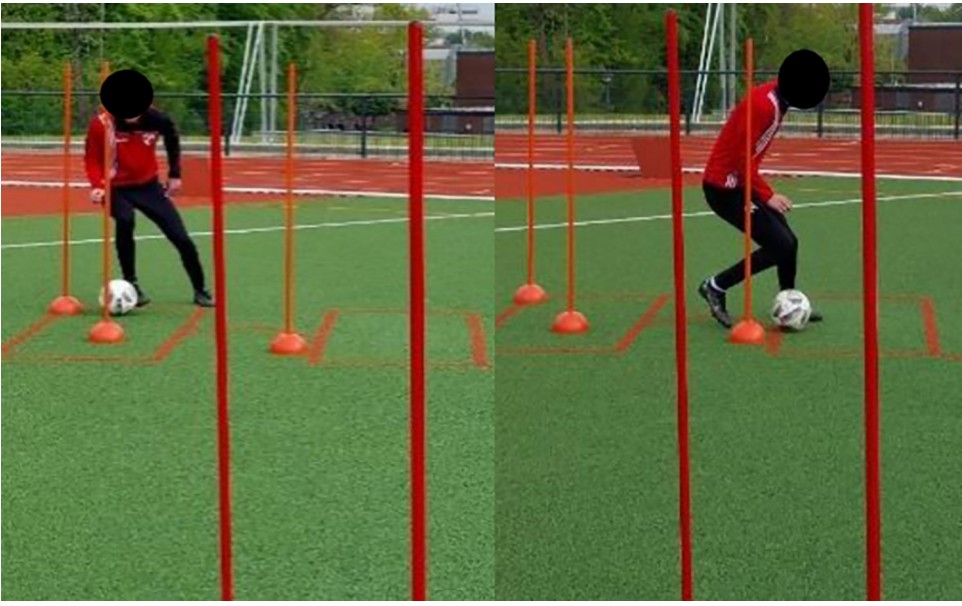

**Fig 4. Dribbling test.**

**Table 1. Mean values, standard deviation (SD) and mean standard error (mean SE) for the 11+ and Football+.**

| | | Overall | | | | Females | | | | Males | | | |
|---|---|---|---|---|---|---|---|---|---|---|---|---|---|
| | | Mean(s) | SD(s) | Mean SE(s) | Cohen's d | Mean(s) | SD(s) | Mean SE | Cohens's d | Mean(s) | SD(s) | Mean SE | Cohens's d |
| 20 m | 11+ | 3.26 | 0.23 | 0.04 | 0.42 | 3.39 | 0.21 | 0.03 | | 3.17 | 0.19 | 0.02 | |
| | Football+ | 3.17 | 0.44 | 0.07 | | 3.36 | 0.20 | | 0.22 | 3.11 | 0.20 | | 0.61 |
| IA | 11+ | 17.68 | 0.95 | 0.15 | 1.43 | 18.31 | 0.82 | 0.11 | | 17.23 | 0.77 | 0.10 | |
| | Football+ | 17.03 | 1.03 | 0.17 | | 17.78 | 0.72 | | 1.24 | 16.48 | 0.87 | | 1.62 |
| DS | 11+ | 14.33 | 3.24 | 0.52 | 0.38 | 16.29 | 3.25 | 0.44 | | 12.90 | 2.42 | 0.26 | |
| | Football+ | 13.72 | 2.53 | 0.41 | | 14.99 | 3.40 | | 0.73 | 12.79 | 2.24 | | 0.09 |
| CMJ | 11+ | 31.20 | 7.60 | 1.23 | -0.13 | 28.39 | 6.56 | 0.95 | -0.27 | 33.98 | 6.76 | 0.57 | |
| | Football+ | 31.63 | 7.21 | 1.17 | | 27.36 | 7.16 | | | 33.99 | | 6.86 | -0.003 |

SD = standard deviation, SE = standard Error

the outcomes of Ayala et al. are highly comparable with the current study. Meanwhile, that study suffers from a tightly limited sample (n = 16) and potentially underpowered results [15]. Additionally, Cloak et al. (2014) compared the acute effects of the 11+ alone with a combination of the 11+ and acute vibration training and found no difference in agility among collegiate players [18]. On the other hand, Bizzini et al. (2013), by investigating 20 male amateur players, found that the 11+ could be an appropriate warm-up, optimizing agility, sprinting, and vertical jump [16]. However, that study did not compare the 11+ to any other warm-up modality and only compared the outcomes with the literature and a resting control condition. That this study undertook a male sample and the current study consisted of a mixed sample could be another reason to testify such inconsistency [16].

There have been several studies assessing the effects of different warm-ups on acute performance, the majority of which reported the applicability of dynamic and intensive warm-ups to properly prepare athletes for high-level performance [22, 23, 37–39]. For example, Abade et al. (2017) suggested that plyometric and repeated changes of direction appear to be useful warm-up exercises to optimize sprinting and jump performance [39]. Hence, improvements in agility and sprinting following application of the Football+ could be due to highly intensive plyometric and anaerobic exercises and cutting maneuvers. Additionally, Zois et al. (2015) showed that small-sided games during warm-up optimize subsequent performance compared with team-sport warm-up [22]. Thus, improvements in the dribbling skill could be due to the small-sided games integrated into the Football+ program. Conducting a dynamic warm-up while benefiting from ball-based exercises may not only prepare players right in the soccer context but also link the warm-up strategy to soccer-specific demands. That the 11+ produces no effect on technical skills appears to be very realistic given that no soccer-specific drill exists in the program.

It has been demonstrated that walking lunges and plyometric exercises improve jump height [27, 39]. Additionally, it has been proven that the 11+ increases quadriceps and hamstring strength, which can contribute to enhancement of lower extremity power in soccer players [7]. The nonsignificant difference in CMJ can be explained by the fact that both warm-ups included that exercise, although the dosage of plyometric exercises in the Football+ group was nearly twice that in the 11+ group. On the other hand, the 11+ includes squat exercises that might improve jump performance [27, 39]. Tobin et al. (2014) demonstrated that performing a series of plyometrics significantly improves CMJ and could be an efficient method of obtaining advantages of the postactivation potentiation phenomenon (PAP) [40].

The differences in agility and sprinting following application of the 11+ and Football + warm-ups can be discussed based on their structure. The 11+ involves more static than

dynamic exercises, with a focus on strengthening core and hip muscles. Although highly effective in terms of injury prevention, such a static structure, including high volume strength exercises, may not prepare the players for further intensive skilled tasks. Parameters such as agility and sprinting depend on the training features, including coordination, mobility, leg power, and speed [28, 41]. Thus, applying exercises that stimulate those features may optimize them. The volume of anaerobic exercises in the Football+ is approximately twice that in the 11+, which seems to be the main reason for the optimized agility and sprinting. In this regard, Faigenbaum et al. (2005) found that performance in the vertical jump, long jump, and shuttle run decreased by 6.5%, 1.9% and 2.6%, respectively, following a low-intensity aerobic warm-up and concluded that pre-event low intensity exercises might be suboptimal for preparing kids for activities requiring high power [42]. Thomas et al. (2009) indicated that plyometric exercises could improve players' muscular power and agility [43].

According to gender, males outperformed females across all parameters regardless of the warm-ups, which is entirely in line with previous studies [19, 44]. Further, although differences were observed across all parameters, the paired t test failed to reach a statistically significant level for 20-m sprinting and CMJ in females and DS and CMJ in males, which is most likely due to the shrinking sample size determined by power analysis. On the other hand, such a contradictory results might also emphasize that trainers should consider gender-specific characteristics to establish goals about the technical and physical performance profiles of players. Meanwhile, further gender-specific studies with larger samples are required to address the acute effects of both programs on male and female players separately. In this regard, it has been demonstrated that utilization of the same exercise method or implementation of the same training volume for both genders may lead to weak training effects [20].

Although useful in injury prevention, the current findings highlight that the 11+ is not a proper warm-up for competitions and matches, as it may not prepare amateur players for subsequent skilled tasks and optimize their performance. Given that delivery of the 11+ to football administrations has remained challenging, such a lack of efficacy on acute performance potentially reduces the 11+ applicability for being used before competitions and consequently lowers the compliance and implementation of the program. Therefore, fundamental modifications on the 11+ aiming to link performance and injury prevention approaches appear to be intransitive and turn to the center of attention considering that the 11+ has not been updated since its launch in 2006. Trainers in amateur football are recommended to apply the Football+ program as a warm-up routine before competitions and high-load training sessions and benefit from its advantage in optimizing motor performance.

## Conclusion

A well-structured football-specific dynamic warm-up including dynamic stretching, roughly intensive running exercises, strengthening, small-sided games, and plyometrics properly optimizes acute performance and improves sprinting, agility, and dribbling compared to the 11+. Although practical in injury prevention, the 11+ may not optimize acute performance properly and should not be performed before competitions and high-demand training sessions. A new framework to incorporate both soccer-specific and injury prevention exercise into the program may justify the applicability of the 11+ as a warm-up routine and may potentially enhance the delivery and compliance of the program. This study provides empirical evidence behind the applicability of the Football+ program before matches and competitions in amateur football. Practitioners and trainers are highly recommended to apply the Football+ program and benefit from its advantage in performance optimization before competitions and matches. Further studies should evaluate the long-term effects of the Football+ on performance and injury prevention.

## Limitations

This study involved no control group, and therefore, we could not compare the effects of two warm-up modalities with traditional warm-ups being used in amateur football. Further, given that the sample size was relatively small, dividing the players based on gender resulted in non-significant and contradictory results. Therefore, further gender-specific studies including larger samples are required to better identify the acute 11+ effects on different genders.

## Supporting information

**S1 Data.**
(XLSX)

**S1 Table. Mean values, standard deviation (SD) and mean standard error (mean SE) for the 11+ and Football+.**
(DOCX)

**S1 Appendix. The Football+ program.**
(DOCX)

## Acknowledgments

The authors would like to express their gratitude to Kevin Nolte for his support in data processing.

## Author Contributions

**Conceptualization:** Mojtaba Asgari.

**Data curation:** Benedikt Terschluse, Maximilian Sueck.

**Formal analysis:** Mojtaba Asgari.

**Methodology:** Mojtaba Asgari.

**Project administration:** Mojtaba Asgari.

**Supervision:** Thomas Jaitner.

**Writing – original draft:** Mojtaba Asgari.

**Writing – review & editing:** Marcus Schmidt, Thomas Jaitner.

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
