## [Decision Letter · Decision Letter 0]

16 Jan 2023

PONE-D-22-35295Acute effects of the FIFA11+ and Football+ warm ups on motor performance. A crossover randomized controlled trial.PLOS ONE

Dear Dr. Asgari,

Thank you for submitting your manuscript to PLOS ONE. After careful consideration, we feel that it has merit but does not fully meet PLOS ONE’s publication criteria as it currently stands. Therefore, we invite you to submit a revised version of the manuscript that addresses the points raised during the review process.

ACADEMIC EDITOR:Dear authors,

The work is interesting but in the current form, there are several issues that need to be addressed before further consideration. Indeed, there are some important issues that may lead to rejection. Even so, I believe that it is more correct and ethical to give a chance for authors to explain.

Thus, I suggest major revisions.

Please address and answer all comments made by the three reviewers.

Thank you

BR

Please submit your revised manuscript by Mar 02 2023 11:59PM. If you will need more time than this to complete your revisions, please reply to this message or contact the journal office at plosone@plos.org. Please include the following items when submitting your revised manuscript:A rebuttal letter that responds to each point raised by the academic editor and reviewer(s). You should upload this letter as a separate file labeled 'Response to Reviewers'.A marked-up copy of your manuscript that highlights changes made to the original version. You should upload this as a separate file labeled 'Revised Manuscript with Track Changes'.An unmarked version of your revised paper without tracked changes. You should upload this as a separate file labeled 'Manuscript'.

We look forward to receiving your revised manuscript.

Kind regards,

Rafael Franco Soares Oliveira

Academic Editor

PLOS ONE

Journal Requirements:

Reviewers' comments:

Reviewer's Responses to Questions

**Comments to the Author**

1. Is the manuscript technically sound, and do the data support the conclusions?

Reviewer #1: Partly

Reviewer #2: Yes

Reviewer #3: Yes

2. Has the statistical analysis been performed appropriately and rigorously? 

Reviewer #1: Yes

Reviewer #2: Yes

Reviewer #3: Yes

3. Have the authors made all data underlying the findings in their manuscript fully available?

Reviewer #1: Yes

Reviewer #2: Yes

Reviewer #3: Yes

4. Is the manuscript presented in an intelligible fashion and written in standard English?

Reviewer #1: Yes

Reviewer #2: Yes

Reviewer #3: Yes

5. Review Comments to the Author

Reviewer #1: Thank you for the opportunity to review this article. The paper addresses a novel under-researched area, which has the potential to provide useful recommendations for coaches. However, there are some questions that need to be addressed to the manuscript.

Specific comments are provided below:

ABSTRACT

Remove “SPSS V.29”

KEYWORDS

Change the keywords that match in the title (line 26)

INTRODUCTION

His study focuses on men and women. However, the introduction does not distinguish between the two genders. Please clarify this.

Also, please update the references and include the results of meta-analyses related to this topic.

Male or female? (sentence 2, pag 3)

Add references (sentence 4, pag 3)

Add results of all studies included in the second paragraph

METHODS

Participants

Why four participants dropped out during the study period? Add it

Performance tests

What kind of boots did you use the participants during performance tests?

Include the height of time gates

Statistical analyses

Please divide the sample by gender and perform the statistical analysis.

The second paragraph of this sections should be in the results section. Change it.

RESULTS

Include Cohen's d values in table 1.

DISCUSSION

Add a paragraph with the limits of the study and possible future studies.

CONCLUSIONS

There is not a proper conclusion where the main findings are highlighted. Add guidelines on how relevant are the findings of your study for physical trainer

APPENDIX 1.

Is it possible to represent each exercise as a figure? That would help the reader to understand it better

Reviewer #2: First, I would like to congratulate the authors for the pertinence of the study. I will, however, present some suggestions for improvement:

- Abstract must comply with the rules of the journal

- Introduction: Refer what the present study adds to existing knowledge. The objectives must be clarified. According to what exists, is it not possible to establish hypotheses?

Methodology: Interventions - I suggest pairing an illustration or table with the programs used.

Results: Notes should be placed on abbreviations in the table. Improve the table so that all results are clearly displayed.

Discussion: I suggest that results are not presented in the discussion. Authors should mention the limitations of the study and propose future investigations.

Present the practical implications of the study.

Review the formatting of references.

Reviewer #3: Congratulations to the authors. The study has a simple method and findings section. The method section is not written in detail. Given the quality of the journal (my quarterly tranche Q1), I would like to reject the study.

- The data collection process cannot be detailed. Where, by whom, and in what time of day the data (mornings???) were collected. It should be detailed. How was it applied to a total of 48 people? On different days? If 48 people practiced in the same session, waiting time during different tests may reduce the warming effect. What action was taken for this? The Experimental design should be illustrated?

-11+ exercises are mentioned in the Interventions section. Like the Football+ programme, it should be presented in detail in the table.

-What kind of advice was given to the athletes before the measurements (sleep, nutrition). should be explained.

Why were men and women evaluated in the same group? Different tables can be created for both genders.

In the study, two different protocols were tested on the participants. There could have been 3 different protocols in the study. That is, the Study could only have a control group in the form of general warm-up or jogging to determine the effect of other warm-up protocols according to control group.

How the intensity of the exercise was adjusted during the both warm-up protocols in all participants. It should be disclosed.

The discussion section is not detailed enough. Physiological mechanisms have not been adequately explained. Up-to-date resources are required.

The study has a simple method and findings section. The method section is not written in detail.

6. PLOS authors have the option to publish the peer review history of their article (what does this mean?). If published, this will include your full peer review and any attached files.

Reviewer #1: No

Reviewer #2: **Yes: **Fernando Santos

Reviewer #3: **Yes: **Halil İbrahim Ceylan

---

## [Author Response · Author response to Decision Letter 0]

16 Mar 2023

Point-to-point Responses to reviewers

ACADEMIC EDITOR:

Dear authors,

The work is interesting but in the current form, there are several issues that need to be addressed before further consideration. Indeed, there are some important issues that may lead to rejection. Even so, I believe that it is more correct and ethical to give a chance for authors to explain.

Thus, I suggest major revisions.

Please address and answer all comments made by the three reviewers.

Thank you.

Response to academic editor

Dear academic editor,

On behalf of my research team, I would like to thank you so much for your positive feedback and considering the work for further assessments. We have tried to address all the points and deficiencies identified by the reviewers and believe that the point you and the reviewers raised would absolutely improve the manuscript quality.

Reviewer #1: Thank you for the opportunity to review this article. The paper addresses a novel under-researched area, which has the potential to provide useful recommendations for coaches. However, there are some questions that need to be addressed to the manuscript.

***dear reviewer 1.

On behalf of my research group, I would like to thank you so much for your constructive comments and commentary on our manuscript. We believe that the points you have raised strongly improve the quality of this work. Therefore, our effort was to address all the points and integrate them throughout the manuscript. Revisions made based on your comments are highlighted in yellow throughout the manuscript. 

Specific comments are provided below:

ABSTRACT

Remove “SPSS V.29”

***The relevant sentence was revised accordingly.

KEYWORDS

Change the keywords that match in the title (line 26)

***The keywords were changed accordingly

INTRODUCTION

This study focuses on men and women. However, the introduction does not distinguish between the two genders. Please clarify this.

Additionally, please update the references and include the results of meta-analyses related to this topic.

***Thank you so much for your constructive comments. The references were updated, and additional information was added accordingly. Information related to gender was added to the introduction.

Male or female? (sentence 2, pag 3)

***males 

Add references (sentence 4, pag 3).

***The reference was added.

Add results of all studies included in the second paragraph

*** Thank you very much for your comment. The related results were added to the end of the second paragraph. 

METHODS

Participants

Why four participants dropped out during the study period? Add it.

*** they were excluded due to Coronavirus infection 

Performance tests

What kind of boots did you use the participants during performance tests?

***All players put on the soccer shoes that are regularly used in artificial turf 

Include the height of time gates

*** Time gate height was added

Statistical analyses

Please divide the sample by gender and perform the statistical analysis.

***thank you very much for your recommendation. We performed the statistical analysis based on gender and reported the outcomes accordingly as per your request. However, the power analysis reveals that the sample needed for this study is 34, reducing the sample based on the gender potentially leads to underpowered results, and consequently, non-significant outcomes came out. 

The second paragraph of this sections should be in the results section. Change it.

*** The second paragraph was moved to the result section.

RESULTS

Include Cohen's d values in table 1.

***thank you very much. The values were already mentioned in the text. Meanwhile, they were added to the table as per your request. 

DISCUSSION

Add a paragraph with the limits of the study and possible future studies.

***thank you for your constructive comment. A limitation section was added to the manuscript 

CONCLUSIONS

There is not a proper conclusion where the main findings are highlighted. Add guidelines on how relevant are the findings of your study for physical trainer.

***Thank you so much. The conclusion was revised entirely. 

APPENDIX 1.

Is it possible to represent each exercise as a figure? That would help the reader to understand it better

*** Thank you very much for your comment. Unfortunately, the photos for all exercises are not available currently. However, we are planning to make poster for the Football+ program and will publish it once it becomes available. 

Reviewer #2: First, I would like to congratulate the authors for the pertinence of the study. I will, however, present some suggestions for improvement.

***dear reviewer 2.

On behalf of my research group, I thank you so much for your great feedback and constructive commentary on the manuscript. The points you have raised would absolutely induce the quality of the work, and thus, we tried to integrate them throughout the manuscript. The changes made based on your comments are highlighted in green.

- Abstract must comply with the rules of the journal

*** thank you very much. We checked the abstract format.

- Introduction: Refer what the present study adds to existing knowledge. The objectives must be clarified. According to what exists, is it not possible to establish hypotheses?

Thank you very much. We have integrated your suggestion throughout the introduction and revised it accordingly. 

Methodology: Interventions - I suggest pairing an illustration or table with the programs used.

***thank you very much. Given that the 11+ is a well-known and somewhat commercial programme and its table and catalog have been published several times in different languages, we assumed that citing the related references would suffice. However, given that two reviewers agreed on this point, we added the 11+ description to the text. 

Results: Notes should be placed on abbreviations in the table. Improve the table so that all results are clearly displayed.

***thank you. We added footnotes to the table.

Discussion: I suggest that results are not presented in the discussion.

***thank you. We removed the results from the discussion.

Authors should mention the limitations of the study and propose future investigations.

***thank you. We added a limitation section to the end of the manuscript.

Present the practical implications of the study.

***thank you for your constructive comment. We added a relevant sentence to the end of the discussion.

Review the formatting of references.

***The references were adjusted according to the journal guidelines.

Reviewer #3: Congratulations to the authors. The study has a simple method and findings section. The method section is not written in detail. Given the quality of the journal (my quarterly tranche Q1), I would like to reject the study.

Dear reviewer3.

On behalf of our research team, I thank you so much for your commentary on this work. The critical points you highlighted increase the quality of the work, and hence, we have tried to address them properly throughout the manuscript. The changes we made based on your comments are highlighted in light blue.

- The data collection process cannot be detailed. Where by whom, and at what time of day the data (mornings???) were collected. It should be detailed. How was it applied to a total of 48 people? On different days? If 48 people practiced in the same session, waiting time during different tests may reduce the warming effect. What action was taken for this? The Experimental design should be illustrated?

***Thank you so much for your constructive comment. Additional information was adjusted to the text as per your suggestions. The measurements were conducted in mornings as the training sessions of the players was also at mornings. 

-11+ exercises are mentioned in the Interventions section. Like the Football+ programme, it should be presented in detail in the table.

***thank you very much. Given that the 11+ is a well-known and somewhat commercial programme and its table and catalog have been published several times in different languages, we assumed that citing the related references would suffice. However, given that two reviewers agreed on this point, we added the 11+ description to the text. 

-What kind of advice was given to the athletes before the measurements (sleep, nutrition). should be explained.

***thank you for your feedback. We asked the players to not perform heavy physical activities 48 h before measurements, sleep no later than 12 pm the night before measurements and drink only caffeine-free liquids 10 h before measurements. The suggestions were added to the text. 

Why were men and women evaluated in the same group? Different tables can be created for both genders.

*** Please divide the sample by gender and perform the statistical analysis.

***thank you very much for your recommendation. We have performed statistical analysis based on the gender and reported the outcomes accordingly.

In the study, two different protocols were tested on the participants. There could have been 3 different protocols in the study. That is, the study could only have a control group in the form of general warm-up or jogging to determine the effect of other warm-up protocols according to the control group.

***thank you for your constructive recommendation. This study focused on the acute effects of the 11+ as a gold standard for injury prevention in comparison to a football-specific warm-up, the so-called football+. Given that the previous studies were also compared the 11+ to a dynamic warm up, we assumed that having two structured warm up modalities including different contents might be more practical than having a control group alongside the 11+.

How the intensity of the exercise was adjusted during both warm-up protocols in all participants. It should be disclosed.

***two practitioners who have been doing the 11+ programme for up to 2 years were adjusted the training intensity by asking the players to keep the intensity level as high as the practitioners themselves do. Generally, in this kind of interventions, it is quite common to adjust the intensity based on individual‘s estimations simply because there is no other alternative to control that. The 11+ itself also uses self-estimated approach to control the intensity of the exercises.

The discussion section is not detailed enough. Physiological mechanisms have not been adequately explained. Up-to-date resources are needed.

***thank you for your constructive comment. We revised the discussion and integrated additional inputs. References were updated as per your request.

---

## [Decision Letter · Decision Letter 1]

3 Apr 2023

PONE-D-22-35295R1Acute effects of the FIFA11+ and Football+ warm ups on motor performance. A crossover randomized controlled trial.PLOS ONE

Dear Dr. Asgari,

Thank you for submitting your manuscript to PLOS ONE. After careful consideration, we feel that it has merit but does not fully meet PLOS ONE’s publication criteria as it currently stands. Therefore, we invite you to submit a revised version of the manuscript that addresses the points raised during the review process.

ACADEMIC EDITOR:Dear authors,

Congratulations on the work already performed. Reviewers already recommended acceptance of your work, and I just have some minor suggestions to improve some parts.

If you address them, the paper will be accepted in the next round:

-in the 1st line abstract, authors stated that: "Few studies including small samples have addressed the acute effects of the 11+ on motor performance." However, I believe this sentence needs to change because in your limitations, authors also stated that "given that the sample size was relatively small, dividing the players based on gender resulted in nonsignificant and contradictory results." Therefore, please change the abstract;

-authors also mentioned in abstract that "Contradictory results were found according to gender." This is not clear enough. Please be more precise. Authors should have in mind that abstract is like the cover letter of your work. Thus, it must be carefully written;

-finally, in the hypothesis of the study, authors wrote that "Given the roughly intensive, soccer-specific and dynamic structure of the Football+ program, we assume that it results in superior acute performance compared to the 11+." This is also not clear enough. Therefore, please change and support your hypothesis with references.

Thank you

Please submit your revised manuscript by 8 of April. If you will need more time than this to complete your revisions, please reply to this message or contact the journal office at plosone@plos.org. Please include the following items when submitting your revised manuscript:A rebuttal letter that responds to each point raised by the academic editor and reviewer(s). You should upload this letter as a separate file labeled 'Response to Reviewers'.A marked-up copy of your manuscript that highlights changes made to the original version. You should upload this as a separate file labeled 'Revised Manuscript with Track Changes'.An unmarked version of your revised paper without tracked changes. You should upload this as a separate file labeled 'Manuscript'.If applicable, we recommend that you deposit your laboratory protocols in protocols.io to enhance the reproducibility of your results. Protocols.io assigns your protocol its own identifier (DOI) so that it can be cited independently in the future. For instructions see: https://journals.plos.org/plosone/s/submission-guidelines#loc-laboratory-protocols. Additionally, PLOS ONE offers an option for publishing peer-reviewed Lab Protocol articles, which describe protocols hosted on protocols.io. Read more information on sharing protocols at https://plos.org/protocols?utm_medium=editorial-email&utm_source=authorletters&utm_campaign=protocols.

We look forward to receiving your revised manuscript.

Kind regards,

Rafael Franco Soares Oliveira

Academic Editor

PLOS ONE

Journal Requirements:

Additional Editor Comments (if provided):

Dear authors,

Congratulations on the work already performed. Reviewers already recommended acceptance of your work, and I just have some minor suggestions to improve some parts.

If you address them, the paper will be accepted in the next round:

-in the 1st line abstract, authors stated that: "Few studies including small samples have addressed the acute effects of the 11+ on motor performance." However, I believe this sentence needs to change because in your limitations, authors also stated that "given that the sample size was relatively small, dividing the players based on gender resulted in nonsignificant and contradictory results." Therefore, please change the abstract;

-authors also mentioned in abstract that "Contradictory results were found according to gender." This is not clear enough. Please be more precise. Authors should have in mind that abstract is like the cover letter of your work. Thus, it must be carefully written;

-finally, in the hypothesis of the study, authors wrote that "Given the roughly intensive, soccer-specific and dynamic structure of the Football+ program, we assume that it results in superior acute performance compared to the 11+." This is also not clear enough. Therefore, please change and support your hypothesis with references.

Thank you

Reviewers' comments:

Reviewer's Responses to Questions

**Comments to the Author**

1. If the authors have adequately addressed your comments raised in a previous round of review and you feel that this manuscript is now acceptable for publication, you may indicate that here to bypass the “Comments to the Author” section, enter your conflict of interest statement in the “Confidential to Editor” section, and submit your "Accept" recommendation.

Reviewer #1: All comments have been addressed

Reviewer #2: All comments have been addressed

Reviewer #3: All comments have been addressed

2. Is the manuscript technically sound, and do the data support the conclusions?

Reviewer #1: Yes

Reviewer #2: Yes

Reviewer #3: Yes

3. Has the statistical analysis been performed appropriately and rigorously? 

Reviewer #1: Yes

Reviewer #2: Yes

Reviewer #3: Yes

4. Have the authors made all data underlying the findings in their manuscript fully available?

Reviewer #1: Yes

Reviewer #2: Yes

Reviewer #3: Yes

5. Is the manuscript presented in an intelligible fashion and written in standard English?

Reviewer #1: Yes

Reviewer #2: Yes

Reviewer #3: Yes

6. Review Comments to the Author

Reviewer #1: The authors have taken my contributions into consideration in order to improve the level of the manuscript, which is interesting.

Reviewer #2: The authors responded appropriately to the reviewer's comments, and the article is ready for publication.

Reviewer #3: I examined the file. Authors made all revisions. Congratulations. The article is ready for publication.

7. PLOS authors have the option to publish the peer review history of their article (what does this mean?). If published, this will include your full peer review and any attached files.

Reviewer #1: **Yes: **Elena Mainer-Pardos

Reviewer #2: **Yes: **Fernando Santos

Reviewer #3: **Yes: **Halil İbrahim Ceylan

---

## [Editor Report · Decision Letter 2]

6 Apr 2023

Acute effects of the FIFA11+ and Football+ warm ups on motor performance. A crossover randomized controlled trial.

PONE-D-22-35295R2

Dear Dr. Asgari,

We’re pleased to inform you that your manuscript has been judged scientifically suitable for publication and will be formally accepted for publication once it meets all outstanding technical requirements.

Kind regards,

Rafael Franco Soares Oliveira

Academic Editor

PLOS ONE

Additional Editor Comments (optional):

Dear authors,

Thank you for following all suggestions. My recommendation is to accept your work.

Congratulations!

Best regards
---

## [Editor Report · Acceptance letter]

11 Apr 2023

PONE-D-22-35295R2 

Acute effects of the FIFA11+ and Football+ warm-ups on motor performance. A crossover randomized controlled trial. 

Dear Dr. Asgari:

I'm pleased to inform you that your manuscript has been deemed suitable for publication in PLOS ONE. Congratulations! Your manuscript is now with our production department. 

Kind regards, 

on behalf of

Prof Rafael Franco Soares Oliveira 

Academic Editor

PLOS ONE